# Sphingosine Kinase 1/S1P Signaling Contributes to Pulmonary Fibrosis by Activating Hippo/YAP Pathway and Mitochondrial Reactive Oxygen Species in Lung Fibroblasts

**DOI:** 10.3390/ijms21062064

**Published:** 2020-03-17

**Authors:** Long Shuang Huang, Tara Sudhadevi, Panfeng Fu, Prasanth-Kumar Punathil-Kannan, David Lenin Ebenezer, Ramaswamy Ramchandran, Vijay Putherickal, Paul Cheresh, Guofei Zhou, Alison W. Ha, Anantha Harijith, David W. Kamp, Viswanathan Natarajan

**Affiliations:** 1Departments of Pharmacology, University of Illinois at Chicago, Chicago, IL 60612, USA; lhuang82@uic.edu (L.S.H.); panfengfu@hotmail.com (P.F.); prasanthkumarpk@gmail.com (P.-K.P.-K.); debene2@uic.edu (D.L.E.); ramchan@uic.edu (R.R.); vputhe4@uic.edu (V.P.); 2Departments of Pediatrics, University of Illinois at Chicago, Chicago, IL 60612, USA; taras@uic.edu (T.S.); guofei.zhou@nih.gov (G.Z.); harijith@uic.edu (A.H.); 3Department of Medicine, Division of Pulmonary & Critical care Medicine, Jesse Brown VA Medical Center, Chicago, IL 60612, USA; p-cheresh@northwestern.edu (P.C.); d-kamp@northwestern.edu (D.W.K.); 4Department of Medicine, Northwestern University Feinberg School of Medicine, Chicago, IL 60611, USA; 5Departments of Biochemistry and Molecular genetics, University of Illinois at Chicago, Chicago, IL 60612, USA; aha6@uic.edu; 6Departments of Medicine, University of Illinois at Chicago, Chicago, IL 60612, USA

**Keywords:** lung fibroblast, YAP signaling, SPHK1, S1P, TGF-β, pulmonary fibrosis, BLM

## Abstract

The sphingosine kinase 1 (SPHK1)/sphingosine–1–phosphate (S1P) signaling axis is emerging as a key player in the development of idiopathic pulmonary fibrosis (IPF) and bleomycin (BLM)-induced lung fibrosis in mice. Recent evidence implicates the involvement of the Hippo/Yes-associated protein (YAP) 1 pathway in lung diseases, including IPF, but its plausible link to the SPHK1/S1P signaling pathway is unclear. Herein, we demonstrate the increased co-localization of YAP1 with the fibroblast marker FSP1 in the lung fibroblasts of BLM-challenged mice, and the genetic deletion of *Sphk1* in mouse lung fibroblasts (MLFs) reduced YAP1 localization in fibrotic foci. The PF543 inhibition of SPHK1 activity in mice attenuated YAP1 co-localization with FSP1 in lung fibroblasts. In vitro, TGF-β stimulated YAP1 translocation to the nucleus in primary MLFs, and the deletion of *Sphk1* or inhibition with PF543 attenuated TGF-β-mediated YAP1 nuclear localization. Moreover, the PF543 inhibition of SPHK1, or the verteporfin inhibition of YAP1, decreased the TGF-β- or BLM-induced mitochondrial reactive oxygen species (mtROS) in human lung fibroblasts (HLFs) and the expression of fibronectin (FN) and alpha-smooth muscle actin (α-SMA). Furthermore, scavenging mtROS with MitoTEMPO attenuated the TGF-β-induced expression of FN and α-SMA. The addition of the S1P antibody to HLFs reduced TGF-β- or S1P-mediated YAP1 activation, mtROS, and the expression of FN and α-SMA. These results suggest a role for SPHK1/S1P signaling in TGF-β-induced YAP1 activation and mtROS generation, resulting in fibroblast activation, a critical driver of pulmonary fibrosis.

## 1. Introduction

Idiopathic pulmonary fibrosis (IPF) is a chronic and progressive interstitial lung disease of unknown etiology that is characterized by the inflammation of the pulmonary interstitium, the deposition of extracellular matrix proteins, and fibrosis that destroys the alveolar architecture [1,2,3]. While two FDA approved drugs for the treatment of pulmonary fibrosis (PF) slow the disease’s progression, neither is curative [4,5]. Lung transplantation is an option for a minority of patients [6], but the outcomes after the lung transplant are far worse than for other organs [7]. Thus, there is an urgent, unmet need to identify new targets and novel therapies to ameliorate lung fibrosis.

There is evidence for the involvement of bioactive lipid mediators, such as prostaglandin (PGE2) [8,9,10], lysophosphatidic acid [11,12,13], and its G-protein coupled receptors in the pathophysiology of IPF and experimental pulmonary fibrosis [14,15]. Recent evidence strongly suggests a role for sphingolipids—specifically S1P signaling and homeostasis—in the pathogenesis of IPF, as well as animal models of PF [12,16,17,18]. In mammalian cells, S1P is primarily generated by the phosphorylation of sphingosine catalyzed by sphingosine kinase (SPHK) 1 and 2 [19,20,21] and degraded by S1P lyase (S1PL) [22,23,24], S1P phosphatases, and lipid phosphate phosphatases [25,26]. We have recently demonstrated an enhanced expression of SPHK1 and S1PL in the lung tissues of human IPF patients, and there was a direct correlation between an increased expression of SPHK1 and a reduced survival rate in IPF [18,23,26]. Furthermore, a murine model of bleomycin (BLM)-induced pulmonary fibrosis in *Sphk1*-deficient mice confirmed protection against PF, confirming the data from IPF patients depicting an enhanced SPHK1 and S1PL expression in lung tissues [26]. The genetic deletion of *Sphk2* had no impact on BLM-induced lung inflammation and the development of PF in mice [26].

The pathogenesis of IPF and experimental PF is not well understood; however, recent studies suggest the involvement of both immune and non-immune cells in the development and progression of lung fibrosis [27,28]. Among several lung cell types, the alveolar epithelial cells (AECs), fibroblasts, and macrophages have been implicated in PF [29,30,31]. While the importance of SPHK1 in IPF and BLM-induced PF is clear, the precise contribution of SPHK1 from each of the cell types in the pathogenesis of PF and the mechanism(s) of the S1P-mediated development of PF in animal models is unclear. Here, we show that the conditional deletion of *Sphk1* in AECs and fibroblasts (but not in endothelial cells) protected the mice from BLM-induced lung fibrosis. Furthermore, the conditional deletion of *Sphk1* in fibroblasts reduced BLM-induced Hippo/Yes-associated protein (YAP) 1 expression and the inhibition of SPHK1 activity by PF543, attenuated TGF-β-mediated YAP1 expression, as well as the expression of fibronectin (FN) and α-smooth muscle actin (α-SMA) in lung fibroblasts from wild type mice. PF543 treatment of human lung fibroblasts (HLFs) also attenuated BLM- or TGF-β-mediated mitochondrial reactive oxygen species (mtROS) and the inhibition of YAP1, or the knockdown of *YAP1,* reduced mtROS and the expression of FN and α-SMA. These results reveal that the SPHK1/S1P signaling axis in lung fibroblasts regulates BLM- or TGF-β-induced mtROS and the expression of FN and α-SMA through the YAP1 pathway.

## 2. Results

### 2.1. Genetic Deletion of Sphk1 in Fibroblasts and Alveolar Epithelial Cells Protects Mice against Bleomycin-Induced Lung Fibrosis

We reported previously that SPHK1 is upregulated in the lung tissues of IPF patients and BLM-challenged mice [26]. Furthermore, the whole body knockdown of *Sphk1* or the inhibition of SPHK1 with SKI-II, a non-specific inhibitor of SPHK1 and SPHK2 attenuated mortality and PF in mice [26]. To further characterize the relative importance of fibroblast and the lung epithelial cell SPHK1 in the development of PF, we generated conditional knockouts of *Sphk1* in fibroblasts, epithelial cell (Ep), and endothelial cell (EC) by breeding fibroblast-specific protein 1 (*FSP1)-Cre*, surfactant protein C (*SPC)-Cre,* and tyrosine-protein kinase receptor (*Tie)-Cre* mice with *Sphk1^flox/flox^* mice to generate fibroblast specific *Sphk1 (FB-Sphk1KO; Sphk1^flox/flox^*: *FSP1Cre^+^)*, epithelial cell-specific (*Ep-Sphk1KO*; *Sphk1^flox/flox^*: *SPC Cre^+^*), and EC specific (*EC-Sphk1KO*; *Sphk1^flox/flox^: Tie Cre^+^*) knockout mice, respectively (F2 generation). The conditional deletion of *Sphk1* was achieved by administering tamoxifen to animals for 17 days prior to the BLM challenge. These mice (where cell-specific knockdown was confirmed by immunohistochemistry (IHC) and Western blot), and their littermate controls (*Sphk1^flox/flox^*: *Cre^wt^*) were challenged with BLM (1.5 U/kg in 50 µl sterile phosphate-buffered saline (PBS) solution), and the animals were harvested on day 21 post-challenge. When compared with controls, *FB-Sphk1KO* (Figure 1) and *Ep-Sphk1KO* (Figure 2) were protected against BLM-induced PF. The BLM challenge significantly increased lung injury and collagen deposition, as determined by Masson’s trichrome staining, which were reduced in BLM-treated *Sphk1^flox/flox^*: *FSP1Cre^+^* (Figure 1A,D,E) and *Sphk1^flox/flox^*: *SPC Cre^+^* mice (Figure 2A,D,E). Furthermore, the BLM challenge significantly reduced BAL protein and the total cells in *FB-Sphk1KO* (Figure 1B,C) and *Ep-Sphk1KO* mice (Figure 2B,C) compared to the *Sphk1^foxl/flox^* controls. The levels of transforming growth factor-beta (TGF-β), fibronectin (FN), and alpha-smooth muscle actin (α-SMA) were markedly reduced in *Sphk1^flox/flox^*: *FSP1Cre^+^* mice challenged with BLM (Figure 1F,G,H; Appendix A). However, the exposure of *Sphk1^flox/flox^*: *SPC Cre^+^* mice to BLM showed reduced expression of α-SMA, but not FN or collagen 1A2 (Col1A2), as compared to the *Sphk1^flox/flox^* wild type (*WT)* mice (Figure 2F; Appendix A). In contrast to the conditional knockdown of *Sphk1* in fibroblast and AECs, the conditional knockdown of *Sphk1* in ECs using tamoxifen inducible *Tie-Cre* accentuated the BLM-induced lung injury and fibrosis. The BLM challenge of *Sphk1^flox/flox^*: *Tie-Cre^+^* exhibited higher mortality (Figure 3A), lung injury (Figure 3B), and collagen deposition in the lung (Figure 3C) compared to the control *Sphk1^flox/flox^* mice. These results show that SPHK1 in fibroblasts and AECs are pro-inflammatory and pro-fibrotic, while SPHK1 in lung ECs may be anti-inflammatory and anti-fibrotic.

### 2.2. Genetic Deletion of Sphk1 in Fibroblasts Reduces Bleomycin- and TGF-β-Induced YAP1 Expression

The Hippo pathway plays a key role in cellular proliferation, differentiation, and tissue homeostasis [32,33]. YAP and transcriptional coactivator with PDZ-binding motif (TAZ) are the downstream effectors of Hippo signaling, and recent studies have identified the aberrant expression of YAP/TAZ in lung fibrogenesis [34,35,36]. S1P and sphingosyl- phosphorylcholine are bioactive sphingolipids that activate YAP in mammalian cells [37,38], and S1P stimulates YAP activation and cell proliferation through S1P receptors. Little is known about SPHK1/S1P signaling in fibroblasts that mediate fibrogenesis; therefore, we investigated the role of fibroblast SPHK1 in YAP1 expression and activation using *Sphk1 ^flox/flox^* and *Sphk1 ^flox/flox^*: *FSP-Cre^+^* mice. The mice were challenged with BLM for 7 days and the lung tissues were analyzed for YAP1 expression in fibroblasts by the immunofluorescence and co-localization of YAP1 and FSP1 (fibroblast marker). The conditional deletion of *Sphk1* in fibroblasts reduced BLM-induced YAP expression (green) and FSP1 expression (red) (Appendix A) and the merged image (yellow) of FSP1 (red) and YAP1 (green) (Figure 4A,B). In vitro, the exposure of primary mouse lung fibroblasts (MLFs) isolated from *Sphk1 ^flox/flox^* mice to TGF-β (5 ng/mL) for 24 h showed enhanced YAP1 and α-SMA, which was reduced in *Sphk1*-deficient MLFs (Figure 4C,D; Appendix A). These results show the role of SPHK1 in YAP expression in vivo and in vitro.

### 2.3. SPHK1 Activity is Essential for Bleomycin- and TGF-β-Mediated YAP1 Expression in Lung Fibroblasts

Having established a role for the SPHK1 protein in BLM- and TGF-β-induced YAP1 expression, we next investigated if SPHK1 activity is essential for YAP1 expression in fibroblasts. *Sphk1 ^flox/flox^* mice with a normal expression of SPHK1 were treated with PF543, a specific inhibitor of SPHK1 [39] (1 mg/kg body weight, intraperitoneal (IP)) for 24 h prior to the BLM challenge, and the mice were sacrificed 7 days post-BLM challenge. As shown in Figure 5A,B, the administration of PF543 significantly reduced the BLM-induced expression of YAP1, as visualized by the co-localization of YAP1 (green) and FSP1 (red) in the merged image (yellow). Similarly, the pretreatment of primary MLFs with PF543 (1 μM, 1 h) prior to TGF-β (3 h) exposure resulted in the attenuation of FN, YAP1, and α-SMA expression (Figure 5C,D; Appendix A). Taken together, these data show that SPHK1 activity is required for BLM- and TGF-β-induced YAP1 expression in lung fibroblasts in vivo and in vitro, respectively.

### 2.4. Inhibition of SPHK1 by PF543 Attenuates Bleomycin- and TGF-β-Induced Mitochondrial ROS Generation and Expression of FN and α-SMA in Lung Fibroblasts

Both IPF and experimental animal models of lung fibrosis are characterized by a redox imbalance and an increased production of reactive oxygen species (ROS), mainly derived from NADPH oxidase (NOX) 4 and mitochondria. Recent studies also suggest that dysregulated mtROS production promotes mitochondrial dysfunction that drives lung fibrosis [40]. The hyperoxia-induced activation of SPHK1/S1P signaling stimulates NOX2-mediated ROS generation in human lung ECs [38,41], and SPHK1 is targeted to mitochondria during the early stages of stress and modulates the mitochondria unfolded protein response [42]. However, the role of SPHK1 in FN, α-SMA expression, and mtROS production in lung fibrosis is unknown [31]. The exposure of HLFs to TGF-β (5 ng/mL) for 48 h enhanced the expression of FN and α-SMA, which was attenuated by PF543 (Figure 6A–C; Appendix A). In separate experiments, HLFs were pretreated with PF543 (1µM) for 1 h prior to treatment with BLM (1 U/mL) or TGF-β (5 ng/mL) for 3 h, and mtROS production was determined by incubating the cells with MitoTracker (50 nM) and MitoSOX (1 µM) for an additional 30 min. The cells were examined under a fluorescence microscope and the images were quantified. Both the BLM and TGF-β stimulated mtROS generation in HLFs, which was attenuated by PF543 (Figure 6D,E). Taken together, these results suggest that SPHK1 inhibition reduces BLM- or TGF-β-induced mtROS and the expression of FN and α-SMA in HLFs.

### 2.5. Inhibition of SPHK1 Activity with PF543 Attenuates Bleomycin- and TGF-β-Mediated YAP1 Translocation to The Nucleus

Having demonstrated an enhanced YAP1 expression in mouse lungs after the BLM challenge and in MLFs after TGF-β treatment, we next determined YAP1 activation and translocation to the nucleus. Under basal and unstimulated conditions in confluent cells, YAP1 is phosphorylated and primarily localized in the cytoplasm; however, upon stimulation, YAP1 is dephosphorylated by a protein tyrosine phosphatase PTPN14 [43] and translocated to the cell nucleus, where it functions as a co-transcriptional activator [44]. The treatment of HLFs with BLM or TGF-β for 3 h stimulated YAP1 translocation to the cell nucleus, which was inhibited by PF543 (Figure 7). The individual unmerged and merged images after treatment with TGF-β or BLM, with or without PF543, are shown in the Appendix A. This suggests the role of SPHK1 activity in BLM- or TGF-β-induced YAP1 translocation to the nucleus in HLFs.

### 2.6. Inhibition or Downregulation of YAP1 Reduces TGF-β-Induced Mitochondrial ROS Generation and Expression of FN and α-SMA in Lung Fibroblasts

Having demonstrated that blocking SPHK1 activity attenuates BLM- or TGF-β-mediated mtROS and the expression of FN and α-SMA, we investigated the effect of YAP1 inhibition, or the downregulation of YAP1, with shRNA on mtROS, FN, and α-SMA expression in lung fibroblasts. The HLFs were pre-treated with YAP1 inhibitor verteporfin (1 µM) [45,46] for 1 h prior to the TGF-β (5 ng/mL) challenge for 3 h, and mtROS was determined by MitoSOX. The inhibition of YAP1 by verteporfin reduced the TGF-β-induced mtROS (Figure 8A), suggesting a role for YAP1 in mtROS production. Next, we determined the role of YAP1 in TGF-β-induced FN and α-SMA expression. The HLFs were treated with control, or YAP1 shRNA (100 nM), for 48 h, which reduced YAP1 expression >80% compared to the vector shRNA transfected cells (Figure 8B; Appendix A), and the knockdown of YAP1 attenuated the TGF-β-mediated expression of FN and α-SMA (Figure 8B,C). These results show that YAP1 is upstream and essential for TGF-β-induced mtROS generation and the expression of FN and α-SMA.

### 2.7. Mitochondrial ROS is Essential for TGF-β-Induced Expression of FN and α-SMA in Lung Fibroblasts

Having demonstrated that the enhanced expression of FN and α-SMA by TGF-β was SPHK1 and YAP1-dependent and blocking SPHK1 or YAP1 attenuated TGF-β-mediated mtROS, we next investigated the role of mtROS in FN and α-SMA expression. The pre-treatment of HLFs with MitoTEMPO effectively reduced TGF-β-induced mtROS, as determined by MitoSOX (Figure 9A,B). Furthermore, the pre-treatment of HLFs with MitoTEMPO reduced the TGF-β-mediated FN and α-SMA expression (Figure 9C–E; Appendix A). Collectively, these results show the critical role of mtROS in the TGF-β-induced expression of FN and α-SMA in lung fibroblasts.

### 2.8. TGF-β- and S1P-Induced mtROS Production in HLFs is Attenuated with S1P-Antibody

To address the role of the intracellular or extracellular action of SPHK1-derived S1P, we used a S1P antibody to neutralize S1P in the extracellular milieu to determine mtROS production. The pre-treatment of HLFs with S1P Ab—but not IgG—effectively reduced the TGF-β-induced mtROS, as determined by MitoSOX (Figure 10A,B). Furthermore, the pre-treatment of HLFs with the S1P antibody reduced S1P-mediated mtROS production (Figure 10C,D). Together, these results show a critical role for extracellular S1P in inducing mtROS production and the TGF-β-induced expression of FN and α-SMA in lung fibroblasts.

### 2.9. TGF-β-Induced YAP1 Nuclear Translocation in Lung Fibroblasts is Attenuated by S1P Antibody

Our earlier in vivo study with *Sphk1*-deficient mice [26] and the current in vitro results suggest a role for SPHK1-derived S1P in the BLM- and TGF-β mediated mtROS generation and the expression of FN and α-SMA. Both BLM- and TGF-β stimulated S1P generation [26]; however, it was unclear if S1P signaling was intracellular, extracellular, or both [26]. To address the role of the intracellular or extracellular action of SPHK1-derived S1P, we used a S1P antibody to neutralize S1P in the extracellular milieu to determine if TGF-β mediated YAP1 nuclear translocation. As shown in Figure 11A,B, the addition of the S1P antibody (but not IgG) to the media prior to stimulation with TGF-β resulted in the attenuation of YAP1 nuclear translocation in HLFs. Similarly, the S1P-mediated YAP1 nuclear translocation was reduced by the S1P antibody (Figure 11C,D), demonstrating the efficacy of the S1P antibody to neutralize the added S1P and its signaling. Taken together, these data demonstrate that TGF-β-driven YAP1 translocation requires the transport of intracellular S1P to the outside of the cell and the initiation of signaling in HLFs.

## 3. Discussion

Molecular mechanism(s) defining the role of epithelial cells, endothelial cells, and fibroblasts in mediating dysregulated wound repair in pulmonary fibrosis is incompletely defined. Here, we provide evidence that the SPHK1/S1P signaling pathway in the alveolar epithelial cells and fibroblasts—but not in endothelial cells—is pro-fibrotic in a BLM mouse model of pulmonary fibrosis. Additionally, in an in vitro model of TGF-β-induced fibroblast to myofibroblast activation, we provide evidence for SPHK1/S1P►YAP1►mtROS signaling in actively triggering pro-fibrotic responses, which were attenuated by the inhibition of SPHK1, YAP1, and scavenging mtROS production with MitoTEMPO.

Recent studies from our laboratory and others have provided strong evidence for the SPHK1/S1P signaling axis in the pathogenesis of IPF and experimental lung fibrosis [12,18,26,47,48]. SPHK1 is present in all mammalian cells, including lung cells, and here we show for the first time the specific role of SPHK1 in AECs and lung fibroblasts in the development of BLM-induced PF in mice. The S1P levels were elevated in serum and BAL fluid from IPF patients [49], and lung tissues and BAL fluids from the BLM- and radiation-challenged mice [26,50]. Extracellular S1P acts via its receptors on different types of target cells involved in pro-fibrotic and fibrogenic processes through multiple mechanisms, which include vascular permeability, leukocyte infiltration, fibroblast migration, and fibroblast to myofibroblast differentiation. TGF-β, a key cytokine driving fibrogenesis, is upregulated in IPF lungs and BLM-challenged mouse lungs [51,52,53], and regulates *Sphk1* expression and S1P levels in lung fibroblasts [18,26,47,48,49]. Our results here showed that the deletion of *Sphk1* in AECs and fibroblasts conferred the protection against BLM-induced lung fibrosis in mice (Figure 1 and Figure 2), while the deletion of *Sphk1* in endothelial cells did not protect the mice from BLM-induced lung fibrosis, suggesting a differential role for SPHK1-derived S1P from lung epithelial cells and fibroblasts in fibrogenesis compared to the endothelium. We used *Sphk1^flox/flox^* and *FSP1-Cre* mice to delete *Sphk1* in fibroblasts; however, FSP1 is also expressed in other cell types, including macrophages, and can be induced in epithelial cells undergoing fibrosis. Therefore, in addition to fibroblasts, the knockdown of *Sphk1* in macrophages and epithelial cells may also contribute to BLM-induced pulmonary fibrosis. Interestingly, the deletion of *Sphk1* in fibroblasts reduced the BLM-induced expression of both FN and α-SMA in contrast to the epithelial cells, where *Sphk1* deletion only reduced α-SMA. The mechanism for the differential role of *Sphk1* in the modulation of FN and α-SMA in the two cell types was unclear. SPHK1/S1P signaling was shown to affect myofibroblast lesions by inducing A549 epithelial-mesenchymal transition via S1P_2_ and S1P_3_, which is dependent on TGF-β-Smad3, Rho-Rho kinase, and ROS pathways [49]. Furthermore, the TGF-β induced trans-differentiation of myoblasts to myofibroblasts via the upregulation of the SPHK1/S1P_3_ axis, indicating a dual role for the SPHK1/S1P signaling in fibrogenesis [47,48]. In addition to SPHK1, S1P levels in cells were also regulated by S1P lyase that degrades S1P to hexadecenal and ethanolamine phosphate [22,24]. *SGPL1* expression was enhanced in IPF lungs and lungs from BLM-challenged mice [23], and the overexpression of *SGPL1* in HLF decreased the TGF-β-induced FN and α-SMA expression. Interestingly, the partial deletion of *Sgpl1* (*Sgpl1^+/^*^−^) in mice accentuated BLM-induced pulmonary fibrosis in mice [23], suggesting a balance between TGF-β-mediated SPHK1 and S1P lyase expression in regulating S1P levels and fibrogenesis.

An interesting aspect of this study is that the SPHK1/S1P-driven activation of YAP1 by BLM and TGF-β is a potential pathway in lung fibroblast differentiation. The role of SPHK1 in regulating YAP expression became apparent in *Sphk1*-deficient mice, wherein the genetic deletion of *Sphk1* reduced YAP1 expression in fibroblasts (Figure 4), and the knockdown of *SPHK1* with shRNA in lung fibroblasts attenuated TGF-β-induced YAP expression, as well as the expression of FN and α-SMA (Figure 8B,C). Our results also suggest a potential interaction between the SPHK1/S1P and Hippo/YAP pathways in TGF-β induced fibroblast differentiation, as well as fibrosis. The major components of the Hippo pathway, including YAP and its transcriptional coactivator PDZ-binding motif (TAZ), have been shown to be aberrantly activated in several pathologies, including pulmonary fibrosis [33]. In fibroblasts, ECM stiffness mechanoactivates YAP/TAZ, which stimulates the generation of pro-fibrotic mediators and ECM proteins, resulting in tissue stiffness and a feed-forward loop of fibroblast activation and lung fibrogenesis [54]. Active epithelial Hippo signaling has been identified in IPF [36,55]. YAP activity was enhanced in the lung epithelial cells of IPF patients. Also, in human bronchial epithelial cells, the expression of a constitutively active form of YAP (S127A) increased cell proliferation, migration, and p-S6/p-PI3K expression, and this was blocked by the YAP-TEAD inhibitor, verteporfin. Furthermore, in epithelial cells, the YAP/TAZ pathway is involved in epithelial to mesenchymal transition (EMT) and fibroblast differentiation, wherein the disruption of cell polarity and increased ECM stiffness in fibrotic tissues promotes the proliferation and survival of epithelium [34,56]. While our present results show a role for SPHK1/S1P in YAP expression in fibroblasts and fibroblast differentiation, in human bronchial epithelial cells, the YAP pathway has been identified in enhancing the phosphorylation of S6 and PI3K pathways and cell proliferation and migration [36]. In the current study, we have not identified how SPHK1-generated S1P stimulates YAP1 expression and its translocation to the nucleus; however, the results from the S1P antibody experiment suggest the transport of intracellular S1P generated by TGF-β in lung fibroblasts to act extracellularly via one or more of the yet to be defined S1P_1-5_ receptors and YAP1 translocation to the nucleus (Figure 11). In pulmonary artery smooth muscle cells, S1P induced airway SMC proliferation, migration, and contraction by modulating YAP signaling via S1P_2&3_ [57], while the S1P/S1P_3_ axis promoted aerobic glycolysis via YAP in osteosarcoma [58]. Further studies are necessary to characterize the type of S1P receptors involved in TGF-β-mediated YAP activation and expression in fibroblasts to define the potential interaction between SPHK1/S1P/S1PRs in the Hippo/YAP signaling axis and the regulation of fibroblast differentiation and fibrogenesis.

Another pathway that we explored focused on the role of mtROS in fibroblast differentiation. TGF-β stimulated mtROS in HLF that was dependent on SPHK1 expression and activity, as well as YAP activation. The inhibition of SPHK1 and YAP1, or the down-regulation of SPHK1 and YAP1 expression, reduced TGF-β-mediated mtROS (Figure 6 and Figure 7). Furthermore, scavenging mtROS with MitoTEMPO attenuated the TGF-β-dependent expression of FN and α-SMA, demonstrating a definitive role for mtROS in fibroblast differentiation. Interestingly, the exogenously added S1P antibody reduced the TGF-β-induced mtROS, demonstrating the transport of intracellularly generated S1P by TGF-β in mtROS production (Figure 10). Mitochondria is a major source of ROS in cells [59,60,61] and uncontrolled, excessive ROS generation in the mitochondrial matrix can cause mtDNA damage, cytosol–mitochondrial calcium imbalance, and the uncoupling of the electron transport chain [40,59,60,61,62]. Uncontrolled mtROS causes the epithelial cell damage seen in IPF [63] and asbestos-induced lung fibrosis [57,64,65,66]. mtROS has been shown to be critical for hypoxia-induced alveolar epithelial–mesenchymal transition [67], and TGF-β induces senescence in the lung epithelial cells by the upregulation of mtROS [68], suggesting a key role for it in the alveolar epithelial cell EMT and senescence. Furthermore, mitochondrial catalase overexpressed transgenic mice were protected against Crocidolite asbestos-induced lung fibrosis in part via the prevention of alveolar AEC mtDNA damage [65]. Our results show that TGF-β stimulated mtROS in HLFs, which regulated the expression of FN and α-SMA—the markers of fibroblast to myofibroblast differentiation. Additionally, our results show for the first time that SPHK1/S1P signaling regulated mtROS and FN/α-SMA expression via YAP1 in HLF. It is unclear how YAP1 stimulated mtROS and further studies are required to establish the mechanism(s) of the YAP1-mediated regulation of mtROS. mtROS is generated via the electron transport system [69] and NOX4, which is localized in the mitochondria [70,71,72,73,74,75,76]. NOX4 expression is increased in IPF lung specimens and IPF fibroblasts, and increased NOX4 expression is associated with pro-fibrotic phenotype [77,78]. NOX4 expression is increased in tissues with aging [79,80], and a NOX4 knockdown or inhibition of its activity in aged mice restores the resolution of BLM-induced fibrosis [81,82]. The role of mitochondrial NOX4 regulating mitochondrial function is poorly characterized. It has been shown that macrophage NOX4 augments mtROS production in asbestosis [83], suggesting that NOX4 from other cell types, including macrophages, may play a role in enhancing mtROS in fibroblasts. Furthermore, recent studies suggest a potential interaction between mitochondria localized NOX4 and mitochondrial complex I subunits of the electron transport system in modulating mtROS generation [75,84]. Interestingly, the ATP generated in the mitochondria binds to NOX4 and negatively regulates its activity in normal renal epithelial cells and renal carcinoma [85]; however, ATP production in macrophages was NOX4-dependent and NOX4 mediated macrophage polarization to a profibrotic phenotype [83]. In addition to NOX4, the ROS-induced mtROS release orchestrated by NOX4/NOX2 seemed to drive vascular endothelial growth factor (VEGF) signaling via VEGFR2 in the sustained angiogenesis of human umbilical vein ECs [86]. Our current study does not address the role of mitochondrial electron transport and/or mitochondrial NOX4 in the TGF-β-induced generation of mtROS. Further studies are essential to determine potential interactions between SPHK1/S1P/YAP1 and NOX4 in the modulation of TGF-β mediated mtROS and pulmonary fibrosis.

## 4. Materials and Methods

### 4.1. Reagents

A SPHK1 specific inhibitor (PF543) was obtained from Cayman Chemical Co. (Ann Arbor, MI, USA). The recombinant human TGF-β used for the induction of HLF was procured from Pepro Tech Inc. (Rocky Hill, NJ, USA). Mouse α-SMA was purchased from Sigma–Aldrich (St. Louis, MO, USA). The rabbit anti-FN, TGF-β, collagen 1A2, GAPDH, and actin antibodies were from Santa Cruz Biotechnology Inc, CA, USA. The HRP-linked anti-mouse and anti-rabbit IgG antibodies were obtained from Bio-Rad (Hercules, CA, USA).

### 4.2. Mice

*Sphk1 ^flox/flox^*, *SPC-Cre-ERT*, *Tie2-Cre*, and *FSP1-Cre* mice were ordered from Jackson laboratory. *Sphk1 ^flox/flox^* mice were bred with *SPC-Cre-ERT* mice to generate the *SphK1 f/f SPC-Cre-ERT* mice, which induced the specific epithelial knockout of the SPHK1 gene by the uptake of DOX water (1 g of Dox, 40 g of sucrose in 1L of water) for 2 weeks. *Sphk1 ^flox/flox^* mice were bred with *Tie2-Cre* mice to generate the *SphK1 ^f/f^ Tie2-Cre* mice to generate the specific knockout of SphK1 in endothelial cells. *Sphk1 ^flox/flox^* mice were bred with *FSP1-Cre* mice to generate the *SphK1 ^f/f^ FSP1-Cre* mice to generate the specific knockout of *Sphk1* in fibroblast cells. Age- and gende—matched mice were anesthetized, followed by treatment with either saline or BLM sulfate (1.5 U/kg of body weight, Hospira Inc., Lake Forest, IL, USA) in saline by an intratracheal injection and the animals were sacrificed at days 0, 7, and 21 after the BLM challenge. The bronchoalveolar lavage (BAL) fluid was collected by an intratracheal injection of phosphate-buffered saline (PBS) solution, centrifuged, and the supernatants were processed for protein and cytokine measurement. TGF-β1 ELISA kits were obtained from R&D Systems (Minneapolis, MN, USA). The lung tissues were collected for paraffin embedding and cut into 5 μm sections. Masson’s Trichrome staining and Hematoxylin-Eosin staining were done by the Pathology Core Facility. All experiments conformed to the principles of the Animal Welfare Act and National Institute of Health (NIH) guidelines for the care and use of animals in biomedical research, and all the protocols were approved by the Animal Care Use Committee of the institute.

### 4.3. Histopathological Analysis of Pulmonary Fibrosis

The quantitative fibrotic scale/Ashcroft scale was used for investigating the fibrotic changes in lung tissue [23,26]. Briefly, 20 fields were observed at ×100 within each lung section, and individual fields were assessed for severity and scored. A score from 0 (normal) to 8 (total fibrosis) was then given [23,26], which was then averaged for each lung section. All histological specimens were evaluated independently by two individuals in a blind fashion, as well as by a histopathologist, and finally, the mean of the individual scores were taken as the fibrotic score.

### 4.4. Collagen Content in Lung Tissue

The collagen levels in the lung tissue were analyzed as described earlier [23,26,31]. Briefly, the incised right lung lobe from the mice was homogenized in 5 mL of 0.5 M acetic acid in PBS containing 0.6% pepsin. The obtained extracts were rotated overnight at 4 °C, cleared by centrifugation at 10,000× *g* for 15 min, and the collagen content was measured with the Sircol Collagen Assay kit from Accurate Chemical and Scientific Corp, Westbury, NY, USA. The collagen content was presented as micrograms of acid-soluble collagen.

### 4.5. Isolation of Primary Fibroblasts from Mouse Lungs and Cell Culture

The fibroblasts were isolated from the mouse lung as described earlier [31]. Briefly, the lungs from adult mice (6- to 8-weeks-old) were collected, washed well to remove blood, minced into small pieces, and subjected to enzymatic digestion using collagenase type III and DNase I (Worthington Biochemical, Lakewood, NJ, USA) in a Dulbecco’s Modified Eagle Medium (DMEM) medium containing 5% Fetal Calf Serum for 90 min. The solution was then filtered and centrifuged to obtain the cells, which were then washed and cultured in DMEM containing 10% fetal bovine serum (FBS) for 14 days. Primary human lung fibroblasts were obtained from Lonza (Walkersville, MD, USA). The cells were grown and maintained in 6-well dishes with fibroblast growth medium (Lonza, Walkersville, MD, USA). The cells, when ~80% confluent, were serum-starved for 24 h and then stimulated for the indicated time.

### 4.6. Immunofluorescence Microscopy

The HLF were grown in slide chambers/glass-bottomed dishes for all experiments. The cells were pretreated with respective inhibitors/antibodies prior to stimulation with TGF-β1 (5 ng/mL) for specific time periods. The cells were then fixed with 3.7% paraformaldehyde in PBS for 20 min, permeabilized with 0.25% Triton X-100 for 5 min, and blocked with a blocking buffer containing 2% bovine serum albumin (BSA). The cells were then incubated with primary antibodies (1:200 dilutions in blocker) overnight at 4 °C, followed by three good washes (15 min each) in PBS. The cells were then stained with Alexa Fluor-conjugated secondary antibodies (1:1000 dilution in blocker for an hour; Life Technologies, Grand Island, NY), followed by three 15 min washes. Slides were mounted in mounting media with/without 4’,6-diamidino-2-phenylindole (DAPI) and examined under a Nikon Eclipse TE 2000-S fluorescence microscope (Nikon, Tokyo, Japan) or confocal microscope (Zeiss).

### 4.7. Western Blot

Immunoblot analysis was performed as described previously. Briefly, the cell lysates were prepared in lysis buffer containing protease or phosphatase inhibitors, followed by high speed centrifugation. The supernatant containing protein was then quantified, boiled with a Laemmli sample buffer for 5 min, and used. Twenty µg of the total cell lysate was separated on 10% or gradient SDS-PAGE gels, transferred to nitrocellulose membranes, and blocked with 5% BSA prior to incubation with primary antibodies (1:1000 dilution) overnight at 4 °C, followed by secondary antibodies (1:2000 dilution) for 1 h at room temperature. Blots were developed using the ECL chemiluminescence kit, and the integrated density of the bands were quantified using Image J software (NIH, USA).

### 4.8. Statistical Analysis

All data were expressed as mean ± SD from at least three independent experiments. Statistical analysis was done using a Student’s *t*-test or ANOVA. Values of *p* < 0.05 were considered significant.

## 5. Conclusions

In summary, we have demonstrated that in HLF, the activation of SPHK1/S1P signaling by TGF-β regulates the expression of FN and α-SMA via YAP1 and mtROS (Figure 12). TGF-β increased the YAP1 expression in HLF, which was dependent on SPHK1 expression and activity, and blocking both SPHK1 and YAP1 attenuated mtROS and fibroblast differentiation. The TGF-β-dependent mtROS generation and fibroblast differentiation was blocked by the S1P antibody, suggesting the requirement of extracellular S1P signaling via S1PRs. Together, the current study indicates that targeting SPHK1, YAP1, and mtROS, alone or combined, may provide an attractive therapy for PF.

## Figures and Tables

**Figure 1 ijms-21-02064-f001:**
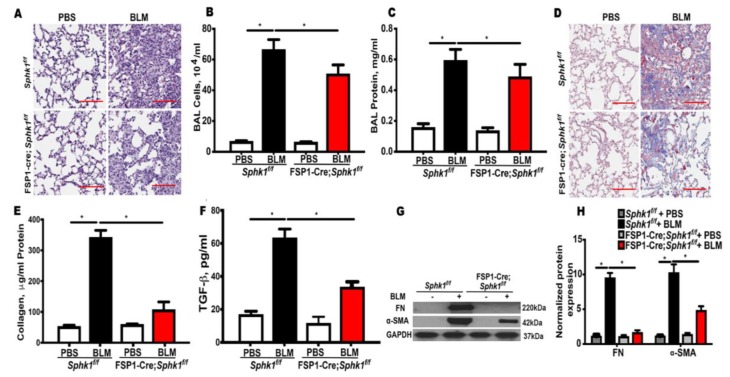
Deletion of *Sphk1* in fibroblasts attenuates bleomycin induced pulmonary fibrosis in mice. *Sphk1^flox/flox^* and *Sphk1^flox/flox^: FSP1Cre^+^* mice (male, 8 weeks) in C57BL/6 background receiving BLM (2 U/kg in 50 µL phosphate-buffered saline (PBS) solution) or PBS intratracheally were sacrificed at day 21 post-challenge. (**A**) Representative H&E images of lung sections from *Sphk1^flox/flox^* and *FB-Sphk1KO* mice with/without BLM challenge. Original magnification, ×10; Scale bar: 200 µm. (**B**) Total cell number in the bronchoalveolar lavage (BAL) fluid. (**C**) Total protein levels in the BAL fluid. (**D**) Representative Masson’s trichrome stains of the lung tissue sections obtained from *Sphk1^flox/flox^* and *FB-Sphk1KO* mice with/without BLM challenge. The histology images show original magnification, ×4; Scale bar: 1 mm. (**E**) Relative quantitative data for acid soluble collagen in lung tissue. (**F**) Relative transforming growth factor beta (TGF-β) levels in the BAL fluids. (**G**,**H**) Protein levels of fibronectin (FN) and α-smooth muscle actin (α-SMA) in lung tissue from *Sphk1^flox/flox^* and *FB-Sphk1KO* mice with/without BLM challenge. Data are expressed as mean ± SEM. * *p* < 0.05; *n* = 4–6 per group.

**Figure 2 ijms-21-02064-f002:**
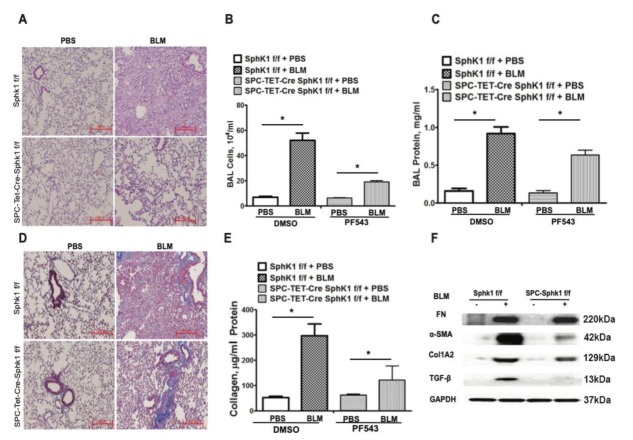
Deletion of *Sphk1* in alveolar epithelial cells protects mice against bleomycin-induced lung fibrosis. *Sphk1^flox/flox^* and *Sphk1^flox/flox^: SPC-Cre^+^* mice (male, 8 weeks; *n* = 4–6 per group) in C57BL/6 background receiving bleomycin (BLM) (2 U/kg in 50 µL PBS) or PBS intratracheally were sacrificed at day 21 post-challenge. (**A**) Illustrative H&E images of lung sections obtained from *Sphk1^flox/flox^* and *SPC-Sphk1KO* mice with/without BLM challenge. Original magnification, ×10; Scale bar: 200 µm. (**B**) Total cell number in bronchoalveolar lavage (BAL), and (**C**) total protein levels in BAL. (**D**) Masson’s trichrome staining for collagen deposition. Representative images of Masson’s trichrome staining of lung sections obtained from *Sphk1^flox/flox^* and *SPC-Sphk1KO* mice with/without BLM challenge (blue arrows showing blue of collagen deposition area). (**E**) Acid soluble collagen levels in lung tissue. (**F**) Expression of fibronectin (FN), and α-smooth muscle actin (α-SMA) in mice lung tissue from *Sphk1^flox/flox^* and *Sphk1^flox/flox^*: *SPC-Cre^+^* mice stimulated with BLM having glyceraldehyde 3-phosphate dehydrogenase (GAPDH) as loading control. * *p* < 0.05 vs. control.

**Figure 3 ijms-21-02064-f003:**
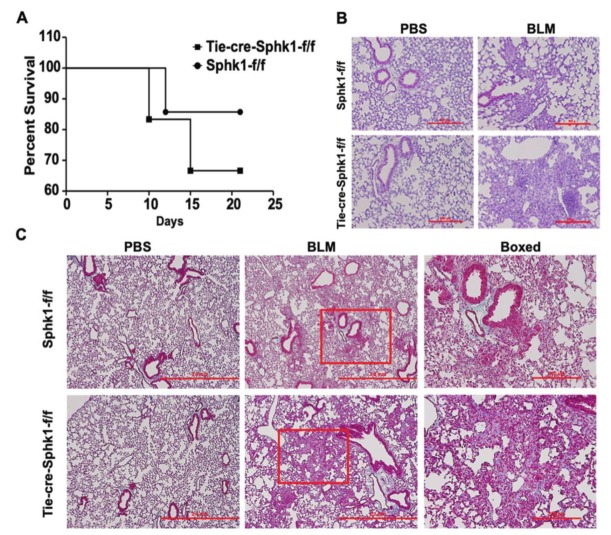
Deletion of *Sphk1* in endothelial cells does not protect mice from bleomycin-induced lung fibrosis. *Sphk1^flox/flox^* and *Sphk1^flox/flox^: Tie-Cre^+^* mice (male, 8 weeks) in C57BL/6 background receiving bleomycin (BLM) (2 U/kg in 50 µl PBS) or PBS intratracheally were sacrificed at day 21 post-challenge. Lungs were removed, embedded in paraffin, and cut into 5 µm sections for staining. (**A**) Survival of *Sphk1^flox/flox^* and *Tie-Sphk1KO* mice challenged with or without BLM challenge. (**B**) Representative H&E photomicrographs of lung sections obtained from *Sphk1^flox/flox^* and *Tie-Sphk1KO* mice with/without BLM challenge. (**C**) Masson’s trichrome staining for collagen deposition. Representative images of Trichrome staining of lung sections obtained from *Sphk1^flox/flox^* and *Tie-Sphk1KO* mice with/without BLM challenge. The area within the red box is zoomed and showed in the boxed panel. Scale bar: 500 µm (Figure B); 2mm and boxed ones 500 µm (Figure C). *n* = 4–6 per group.

**Figure 4 ijms-21-02064-f004:**
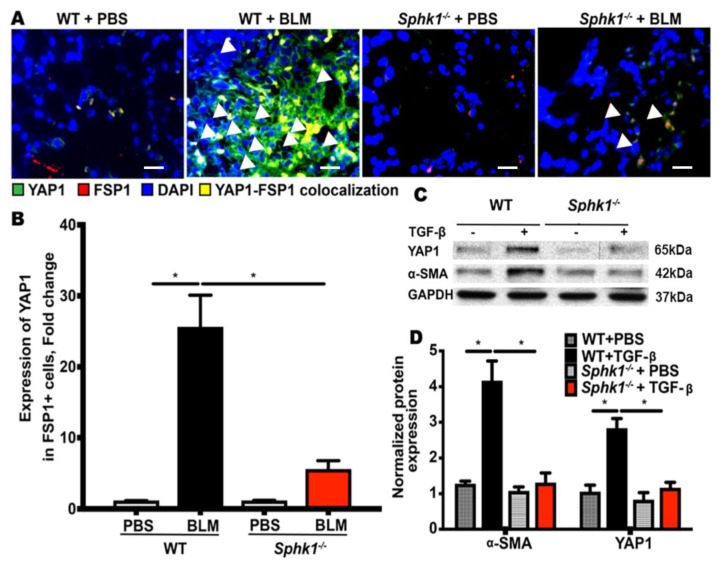
Genetic deletion of *Sphk1* in fibroblasts reduces bleomycin- and TGF-β-induced Hippo/Yes-associated protein (YAP) 1 expression. (**A**,**B**) *Sphk1^flox/flox^* and *Sphk1^flox/flox^: FSP1Cre^+^* mice (male, 8 weeks) in C57BL/6 background receiving bleomycin (BLM) (2 U/kg in 50 µL PBS) or PBS intratracheally were sacrificed at day 21 post-challenge. Lungs were removed, embedded in paraffin, and cut into 5 µm sections for immunofluorescence staining for FSP1 and YAP1. Shown is a representative micrograph of co-localization of FSP1 and YAP1. BLM increased co-localization (yellow) of FSP1 (green) and YAP1 (red) in lung fibroblasts of *Sphk1^flox/flo^*^x^ mice, but not *Sphk1*-deficient mice. The white triangular arrows highlights the YAP1-FSP1 colocalized areas in the image. (**C**,**D**) Lung fibroblasts isolated from *Sphk1^flox/flox^* and *Sphk1*-deficient mice were incubated with transforming growth factor beta (TGF-β) (5 ng/mL) for 48 h and expressions of fibronectin (FN), α-smooth muscle actin (α-SMA), and YAP1 were determined by Western blotting with glyceraldehyde 3-phosphate dehydrogenase (GAPDH) as loading control. TGF-β enhanced YAP1 and α-SMA expression in wild type (WT)mouse lung fibroblasts (MLFs), which was reduced in *Sphk1*-deficient cells. * *p* < 0.05, significantly different from WT cells treated with vehicle, and MLF-deficient in *Sphk1*. *n* = 4–6 per group. Scale bar: 20 µm.

**Figure 5 ijms-21-02064-f005:**
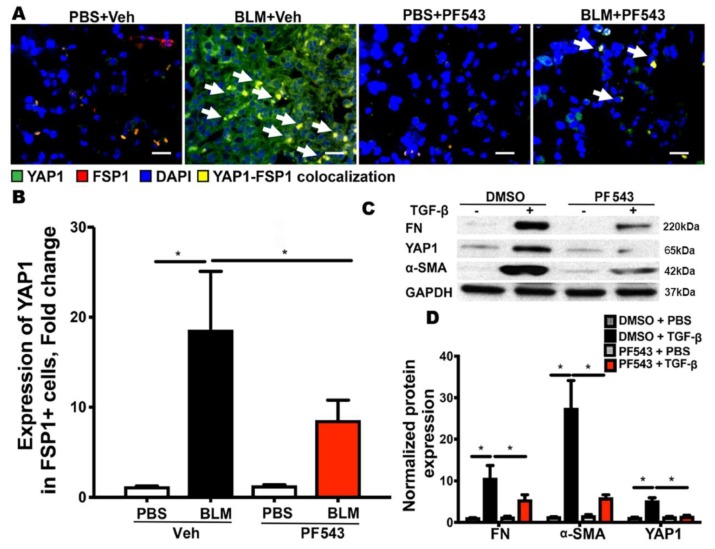
SPHK1 activity is required for BLM and TGF-β-mediated YAP1 expression in fibroblasts. Mice were pre-treated with PF543 on alternate days (1 mg/kg body weight) prior to bleomycin (BLM) challenge and lung tissues were analyzed for expression of fibronectin (FN), Yes-associated protein 1 (YAP1), and α-smooth muscle actin (α-SMA), with GAPDH as loading control. (**A**,**B**) PF543 attenuated YAP1 expression in mouse lungs. The white arrows in (**A**) highlights areas with YAP1-FSP1 colocalization. Scale bar: 20 µm and (**C**,**D**) Expression of FN, YAP1, and α-SMA was reduced by PF543 in mouse lung fibroblasts. * *p* < 0.05 vs. control. *n* = 4–6 per group.

**Figure 6 ijms-21-02064-f006:**
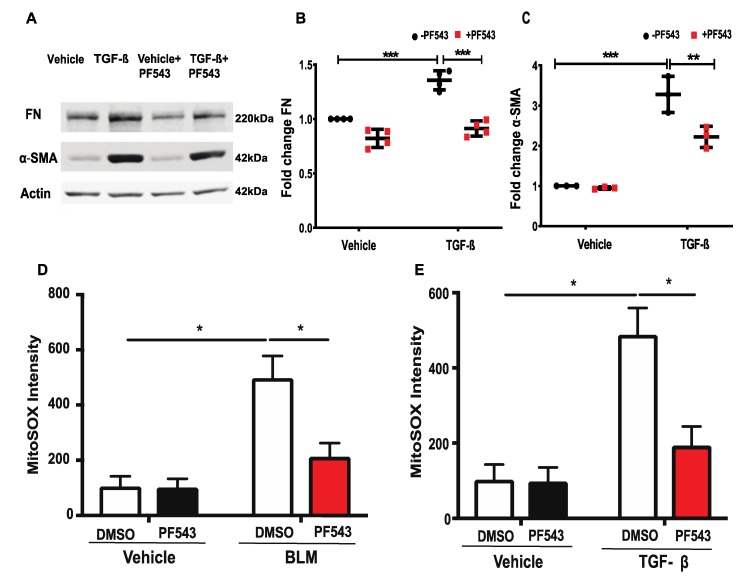
PF543 attenuates TGF-β-induced fibronectin and α-smooth muscle actin protein expression, as well as bleomycin- and TGF-β-induced mitochondrial ROS in lung fibroblasts. (**A**–**C**) Treatment of human lung fibroblasts (HLFs) with transforming growth factor beta (TGF-β) (5 ng/mL) for 48 h increased fibronectin (FN) and α-smooth muscle actin (α-SMA) expression, which was reduced by PF543 (1 µM) treatment. (**D**,**E**) Treatment of HLFs with PF543 (1 µM) for 1 h prior to BLM (1U/mL) or TGF-β challenge for 24 h decreased BLM- or TGF-β-induced mtROS. Data are from three independent experiments in triplicate and values are means + SD. * *p* < 0.05, ** *p* < 0.001, *** *p* < 0.0001 vs. control. *n* = 4.

**Figure 7 ijms-21-02064-f007:**
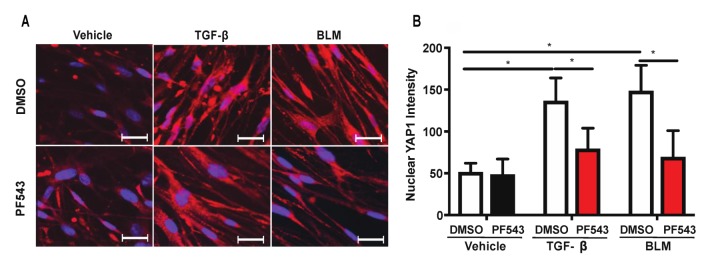
PF543 attenuates TGF-β- and BLM-mediated YAP1 translocation to cell nucleus. (**A**) Human lung fibroblasts (HLFs) grown on glass-bottom 35 mm dishes were treated with PF543 (1 µM) for 1 h prior to TGF-β (5 ng/mL) or BLM (1 U/mL) for 3 h, and YAP1 translocation to cell nucleus was assessed by confocal microscopy. YAP1 is stained red and the nucleus blue (DAPI) and upon nuclear translocation of YAP1, the nucleus is stained pink (red and blue merged). (**B**) Quantitation of (**A**) by Image J analysis showed an increase in YAP1 translocation upon stimulation with TGF-β and BLM. PF543 treatment blocked this nuclear translocation. Seven different areas from three independent dishes were counted for the quantification of nuclear YAP1 with or without PF543 treatment. * *p* < 0.05 vs. control. *n* = 3. Scale bar: 20 µm.

**Figure 8 ijms-21-02064-f008:**
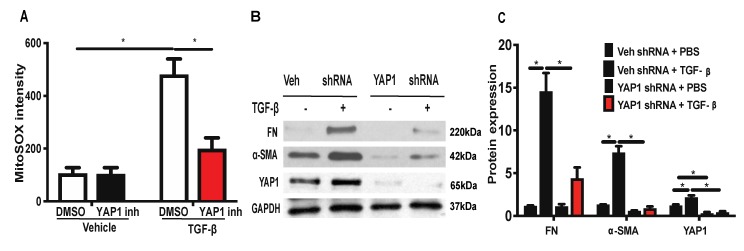
Inhibition or downregulation of YAP1 reduces TGF-β-induced mitochondrial ROS and expression of fibronectin and α-smooth muscle actin. (**A**) HLFs grown on 35 mm glass-bottom dishes were treated with YAP1 inhibitor, verteporfin (1 µM), for 1 h prior to TGF-β (5 ng/mL) challenge for 24 h. After 24 h, cells were treated with MitoSOX for 30 min and mitochondrial ROS (mtROS) was quantified using confocal microscopy. Cells (20–30 numbers) from at least 5–8 different areas of three independent experiments were used for quantification. (**B**) HLFs on 35 mm dishes were transfected with control, or YAP1 shRNA (100 nM), for 48 h to downregulate the YAP1 expression. Cells were challenged with TGF-β (5 ng/mL) for 48 h and cell lysates were analyzed for fibronectin (FN) and α-smooth muscle actin (α-SMA) expression by Western blotting with GAPDH as loading control. Shown is a representative blot from three independent experiments. (**C**) Quantification of (**B**) by densitometry and analysis by Image J. * *p* < 0.05 significantly different compared to control.

**Figure 9 ijms-21-02064-f009:**
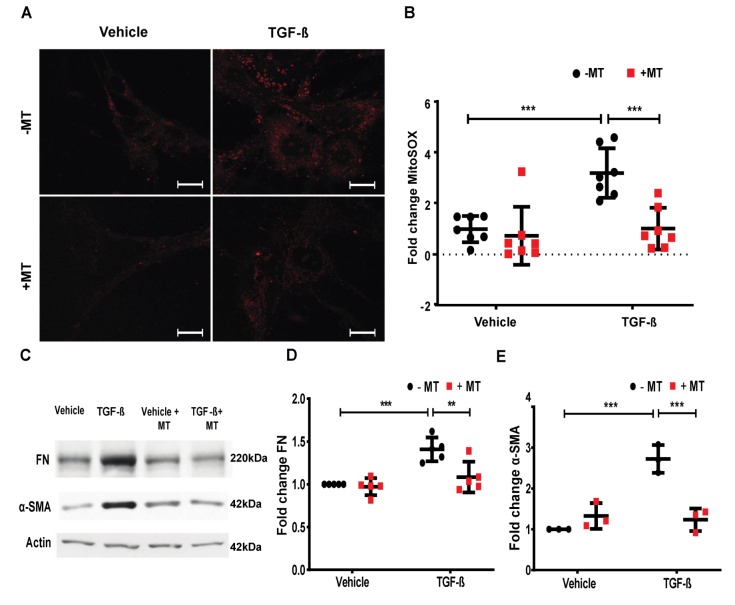
MitoTEMPO reduces TGF-β-induced mitochondrial ROS and expression of fibronectin and α-smooth muscle actin. (**A**,**B**) HLFs grown on 35 mm glass-bottom dishes were treated with the mitochondrial reactive oxygen species (mtROS) scavenger, MitoTEMPO (10 µM), for 1 h prior to transforming growth factor beta (TGF-β) (5 ng/mL) challenge for 24 h. After 24 h, cells were treated with MitoSOX for 30 min and mtROS was quantified using confocal microscopy. Cells (20–30/area) from at least 5–8 different areas of three independent experiments in triplicate were analyzed and quantified. (**C**) HLFs on 35 mm dishes were pre-treated with MitoTEMPO (10 µM) for 1 h prior to treatment with TGF-β (5 ng/mL) for 48 h. Cell lysates were analyzed for fibronectin (FN) and α-smooth muscle actin (α-SMA) expression by Western blotting. Shown is a representative blot from three independent experiments. (**D**) Quantification of FN (**C**), and (**E**) quantification of α-SMA from (**C**) was carried out by densitometry and Image J analysis. ** *p* < 0.001, and *** *p* < 0.0001 significantly different compared to control. MitoTEMPO decreased the TGF-β-induced mtROS, and FN and α-SMA expression. Scale bar: 20 μm.

**Figure 10 ijms-21-02064-f010:**
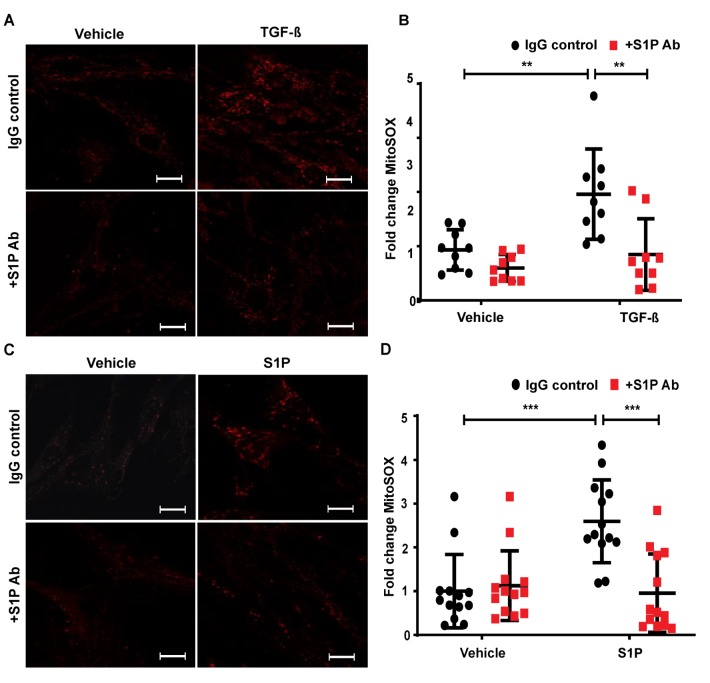
Neutralizing S1P with S1P-antibody attenuates TGF-β- and S1P-mediated mtROS in HLFs. Human lung fibroblasts (HLFs) grown on 35 mm glass-bottom dishes were pre-treated with sphingosine-1-phosphate (S1P) antibody (25 µg/mL) for 1 h prior to TGF-β (5 ng/mL) (**A**,**B**) or S1P (1 µM) (**C**,**D**) challenge for 24 h. After 24 h, cells were treated with MitoSOX for 30 min as described in Methods, and mitochondrial ROS (mtROS) was quantified using confocal microscopy. Cells (20–30 numbers) from at least 5–8 different areas of three independent experiments were used for quantification. ** *p* < 0.001, and *** *p* < 0.0001 as compared to controls. S1P antibody attenuated both TGF-β- and S1P-induced mtROS in HLFs, showing the role of its extracellular signaling via its receptors. Scale bar: 20 μm.

**Figure 11 ijms-21-02064-f011:**
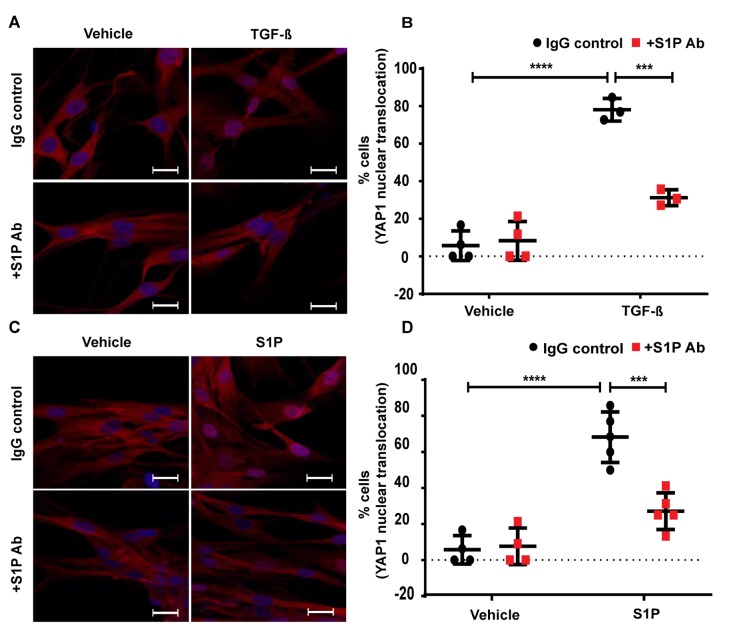
Blocking of S1P with S1P antibody mitigates TGF-β- and S1P- induced YAP1 translocation to the cell nucleus. Human lung fibroblasts (HLFs) grown on 35mm glass-bottom dishes were pre-treated with sphingosine-1-phosphate (S1P) antibody (25 µg/ml) for 1 h prior to TGF-β (5ng/ml) (A,B) or S1P (1 µM) (C,D) challenge for 3 h. (**A**,**B**) The merged images of YAP1 (red) and DAPI (blue) are shown. S1P antibody (S1P Ab) mitigated the TGF-β-induced YAP1 nuclear translocation. (**C**,**D**) S1P antibody also attenuated S1P-induced nuclear translocation of YAP1. *** *p* < 0.001, **** p < 0.0001 as compared to their respective controls. Cells (20–30 numbers) from at least 5–8 different areas of three independent experiments were used for quantification. Scale bar: 20 μm.

**Figure 12 ijms-21-02064-f012:**
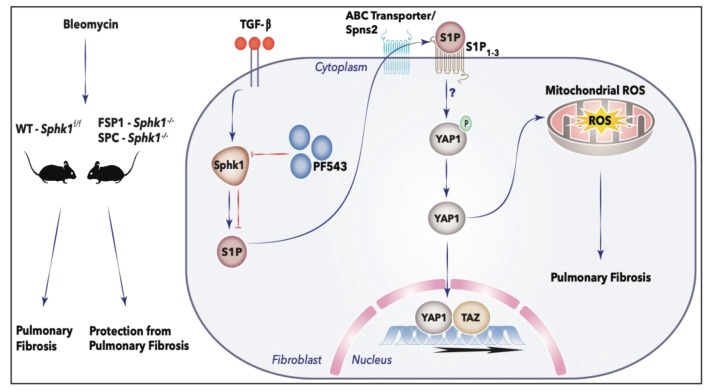
Sphingosine kinase 1/S1P signaling contributes to pulmonary fibrosis by activating Hippo/YAP pathway and mtROS. In vivo, BLM induced pulmonary fibrosis in WT mice, whereas deletion of *Sphk1* in fibroblasts and alveolar epithelial cells conferred protection from pulmonary fibrosis in the Sphk1 knockout mice in the fibroblast or alveolar epithelial cells. In vitro, TGF-β induces activation of SPHK1 in HLFs and blocking SPHK1 activity or expression reduced YAP1 nuclear translocation, mitochondrial ROS (mtROS) production, and expression of fibronectin and α-smooth muscle actin. Experiments with exogenously added S1P-antibody suggest extracellular S1P signaling in YAP1 translocation to nucleus, mtROS generation, and the expression of fibronectin and α-smooth muscle actin. Here, we have not addressed the transport of S1P from inside to outside by S1P transporters and type(s) S1P1-5 receptors involved in transducing the signal inside the cell. The red T bars define inhibition.

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
