# Peer review of "Sphingosine Kinase 1/S1P Signaling Contributes to Pulmonary Fibrosis by Activating Hippo/YAP Pathway and Mitochondrial Reactive Oxygen Species in Lung Fibroblasts"

_ijms, 2020, doi:10.3390/ijms21062064_

Round 1

Reviewer 1 Report

In this interesting study, the authors report that the Sphk1 and S1P are actively involved in bleomycin-inducing pulmonary fibrosis in mice. This is mechanistically mediated by S1P-triggered YAP translocation to the nucleus and increased mtROS formation. By generating inducible fibroblast-, epithelial- and endothelial-specific Sphk1 knockout mice, they further show that particularly fibroblasts and epithelial, but not endothelial cells, are crucial for the fibrosis response.

Major points:

1-Even if FSP-1 was first described as a fibroblast specific protein, it is clear now that other cell types also produce FSP-1. Thus, macrophages were shown as a main source of FSP-1, and in lung fibrosis, FSP-1 is secreted by macrophages (Zhang et al., 2018). Furthermore, FSP-1 can be induced in epithelial cells undergoing EMT. Therefore, the FSP1-Cre-Sphk1 knockout detects a mixed knockout of Sphk1 in various cell types This must be included as a pitfall of the mouse strain.

2-The authors conclude from this study that S1P, by secretion and S1P receptor activation, triggers YAP1 translocation. However, the receptor subtype and the mechanism remains unclear. The authors should at least test the different selective S1P receptor agonists available to get some information about the relevant receptor. 

3-Another conclusion is that YAP1, possibly by upregulation of mitochondrial NOX4, increases mtROS formation, which in turn, contributes to fibrosis.  This would also imply that S1P can increase NOX4 mRNA and/or protein expression. Again, this remains unproven and it is important to address this point (for example by immunostainings of NOX4) to strengthen their hypothesis and the discussion part.

4-In Fig. 2F, the difference in fibronectin between the controls and epithelial-specific knockout of sphk1 is not seen for FN, and hardly seen for col1A2. This needs to be stated more clearly and also explained.

5-In Fig. 4A, the arrows in red are confusing because the color legend indicates FSP1 in red. Instead, the arrows should be in white. FSP1 is hardly detectable although an induction is expected in fibrosis (due to macrophage infiltration and EMT of epithelial cells). Therefore, the FSP1 antibody first must be validated positive in this tissue before reliable conclusions can be made.

6-In Fig. 6A, the western blots of aSMA shows a very strong induction by TGFb, which is not really reduced visibly by PF543.  This increase is not reflected in the evaluation graph Fig. 6C, where  PF543 seems to have a quite complete reducing effect, thus, the blot is not representative. Also the evaluation of the TGFb effect in Fig. 6C is shown as only 50%, whereas the blot shows a huge induction seemingly manifold. The same inconsistency also holds true for blots in Fig. 9C and evaluation. This must be rechecked.

7-Fig. 10A and B only show the effect of S1P-Ab on ROS formation, but not on YAP translocation nor FN or aSMA as stated on lines 191-192. The authors must include these data. Also, experiments with the S1P-Ab lack appropriate controls. Controls show are cells incubated without antibody. However, since IgG is able to activate cells by itself, the appropriate control would be an immunoglobulin in a same concentration. In the legend, please indicate the mol/l of S1P-Ab instead of ug/ml.

8-It is important to show more than just a cut-out band in all the Western blots. The authors must include a molecular weight marker as well for every blot.

Minor points:

9-Pictures of lung sections in Fig. 1A and 1D are too small and hard to judge. Please enlarge.

10-In legend of Fig. 7A, please specify that YAP1 is shown in red, and probably DAPI in blue. Show also a co-localization of YAP1 and DAPI (merged picture).

Author Response

Point-by-Point Response to Reviewer’s Comments

We thank the reviewer for the valuable critique and comments which has contributed to an improved version of the manuscript.

Reviewer #1

We thank the reviewer for providing positive comments and valuable feedback.

 Major points:

1. Thank you for the suggestion. We have added additional sentences highlighting pitfalls of this mouse strain in the discussion section of the revised manuscript. “We used Sphk1flox/flox and FSP-1 Cre mice to delete Sphk1 in fibroblasts; however, FSP-1 is also expressed in other cell types including macrophages and it can be induced in epithelial cells undergoing fibrosis. Therefore, in addition to fibroblasts, Sphk1 in macrophages and epithelial cells may also contribute to bleomycin-induced pulmonary fibrosis”.

2. Thank you for this valuable suggestion. We were allowed only 10 days for resubmission of a revised version. Therefore, not enough time to conduct additional experiments. Further, looking at the receptor subtypes is beyond the scope of this study. We plan to implement additional studies to move forward with the suggestion.   

3. This study shows the involvement of S1P in YAP1 translocation, which possibly increases mtROS via mitochondrial NOX4 and pulmonary fibrosis. Also, S1P by itself could increase NOX4 expression and/or activity, which would contribute to mtROS production. Again, these studies are beyond scope of this manuscript and a separate project is ongoing in our laboratory to determine mechanism(s) of NOX4 activation by S1P and YAP1 in mitochondria.

4. Thanks for pointing out this discrepancy on fibronectin and collagen expression between FSP1 (Figure 1) and SPC (Figure 2) knockout of Sphk1 in fibroblasts and epithelial cells. We have verified the blots from three independent experiments and as depicted in Figure 2, deletion of Sphk1 in alveolar epithelial cells decreased the expression of α-smooth muscle actin (α-SMA) but not fibronectin and collagen 1A2. We have modified the results section to include this differential effect of Sphk1 deletion in fibroblasts and alveolar epithelial cells. The following sentence(s) has been added in the revised manuscript: “The levels of TGF-β, fibronectin (FN) and α-smooth muscle actin (α-SMA) were markedly reduced in Sphk1flox/flox: FSP-1Cre+ mice challenged with bleomycin (Fig. 1 F, G & H).  However, exposure of SphK1flox/flox: SPC Cre+ mice to bleomycin showed reduced expression of α-SMA, but not FN and collagen A2 (ColA2) as compared to the SphK1flox/flox WT mice (Fig. 2 F)”. This is also discussed in the “Discussion” section: “Interestingly, deletion of Sphk1 in fibroblasts reduced bleomycin-induced expression of both FN and α-SMA in contrast to the epithelial cells where Sphk1 deletion only reduced α-SMA. The mechanism for the differential role of Sphk1 in modulation of FN and α-SMA in the two cell types is unclear”.

5. The arrows in red has been changed to white as suggested by the reviewers. The antibody has been validated by the Pathology core for its positivity in this tissue. Also, we have used the validated antibody from the company for which the link is provided below. https://www.abcam.com/s100a4-antibody-epr27612-ab124805.html.

6. The reviewer’s comment is appreciated. We have re-evaluated all the blots from atleast three independent experiments carefully, reanalyzed the bands by densitometry and normalized the data to total actin. In the revised analysis, we noticed ~3.5 fold change in α-SMA compared to vehicle (Fig 6C) and ~2.5 fold increase in Fig 9E. The effect of PF543 on TGF-β-mediated FN and α-SMA expression are statically significant using Student’s t test and ANOVA.

7. Sorry, this was an inadvertent error during conversion of the files to pdf, which was not recognized during the preview of the converted document prior to submission. This has been corrected.  The vehicle in the (-S1P Ab) experiments are the IgG controls. It was named vehicle for easy understanding. In this version, we renamed vehicle as IgG control (Fig. 10) as suggested by the reviewer.

8. As suggested, molecular weight position of the respective bands are included in the revised figures. Due to limited time given for resubmission we are unable to replace the cut-out bands with complete blots   

Minor points:

9. As suggested, Figs 1a and 1D have been enlarged.

10. The images shown are merged ones. The suggested changes have been incorporated. The non-merged individual images followed by merged ones are shown in the supplementary data.

Reviewer 2 Report

This manuscript reveals a potential role for SPHK1/S1P signaling in TGF-b induced YAP1 activity and mitochondrial ROS production and their involvement in pulmonary fibrosis. The results are of interest, but there are issues for the authors to address to improve the overall impact.

1. One of the most critical issues is the causative relationship between PF and other signaling (YAP and mitoROS). Although SPHK1 and PF are correlative, this reviewer feels there is a lack of conclusive evidence between YAP or mtROS and PF. The authors might need to establish better discussion and/or experimental approach whether these events directly or indirectly occur.

2. In terms of logical flow, this reviewer wonders why figure 3 crossed the need to show a series of results presented in figure 1 and 2, e.g. BAL cells, BAL protein, collagen deposition, WB data. This is because, like figure 1 and 2, the authors investigated the effect of BLM-induced lung fibrosis on the deletion of sphk1.

3. The authors should include the number of independent experiments in figure legends. Most of the data lack such information.

4. The paper is difficult to read, and the authors should revise it for better clarity. Also, many errors or mistakes appear in the manuscript. Please correct it carefully.

5. Please unify the reference form shown in the text. (line 45, 50, 53, 59 etc)

6. The square color in Figure 1H does not match the bar graph.

7. Figure 2G & H are missing.

8. No description of figure 1F in the text.

9. Wrong description in figure 3A (line 98). VE-cad Cre+ vs Tie-cre

10. The title of the y-axis in figure 2E is wrong. It should be ug/ml.

11. In figure 4 legend, TFG must be TGF.

12. Description differs from figures. Figure 6A and B show FN, a-SMA expression (line 144)

13. No data for YAP1 in figure 6 (line 146). Also, clarify the description in figure 6C-E (line 146). Description differs from figures.

14. In figure 6A, no information about BLM treatment.

15. In the text, a-SMA is marked differently as aSMA or alpha-SMA, and recommend that the authors unify it. This happens frequently in the text. e.g. TGF-beta or TGF-b.

16. There is no information on scale bar for figure 7a. The authors need to make sure the size of images because the cell/nucleus size look different. It seems very hard to verify that YAP1 nucleus translocation has truly occurred because the DAPI signal is interfering with YAP signal. The authors need better image presentation.

17. In figure 8B, there is no information about ROS generation.

18. Description differs from figures (line 193)

19. line 234, knockdown of sphk1. Figure 8 describes the knockdown of YAP, not sphk1.

20. In figure 10, there is no information about YAP nuclear translocation (line 255)

21. There is no figure 12. It needs correction (line 389)

22. 4. M&M, 4.7 western blot and 4.8. statistical analysis, the font is different, so unify them.

Author Response

We thank the reviewer for the valuable critique and comments which has contributed to an improved version of the manuscript.

Reviewer #2

The reviewer’s comments and suggestions are appreciated.

1. We respectfully disagree with the reviewer’s comment that there is a lack of conclusive evidence between YAP or mtROS and PF. In vitro using lung fibroblasts we have established a causative relationship between SPHK1 and mtROS (Fig. 6) and SPHK1 and YAP1 translocation (activation) (Fig. 7) using PF543. PF543 also reduced TGF-β-induced fibronectin (FN) and α-SMA expression. In Fig. 8, we have demonstrated a role for YAP1 in mtROS production to TGF-beta stimulation and down-regulation of YAP1 or blocking YAP1 activation attenuated expression of α-SMA and FN in human lung fibroblasts. The role of mtROS in TGF-β-mediated FN and α-SMA expression was confirmed using MitoTEMPO, a scavenger of mtROS. These data provide a causative link between SPHK1, YAP1 and mtROS in FN and α-SMA expression modulated by TGF-β.   

2. Deletion of Sphk1 in endothelial using Tie-Cre mice showed no protection against bleomycin-induced pulmonary fibrosis and lung inflammatory injury in mice (Fig. 3). However, deletion of Sphk1 in fibroblasts and alveolar epithelial cells offered significant protection against bleomycin-induced lung fibrosis. Hence, in animals where Sphk1 was knocked down in endothelial cells, we did not measure BAL protein, cytokine levels and cell infiltration.  

3. The number of independent experiments is included in the figure legend of the revised manuscript.

4. Our apologies for the writing style. The revised manuscript has been thoroughly read and edited for errors and mistakes to the best our knowledge. 

5. The references were inserted using the Zotero program as stipulated for IJMS.

6. The necessary changes have been incorporated in Fig. 1H of the revised manuscript.

7. There are no Fig.2 G and H and the text has been revised.

8.  We thank the reviewer for pointing this out. The missed Fig. 1F description is added to the text. 

9. Thanks. VE-cadherin has been corrected to Tie-Cre

10. It was an inadvertent error and thanks for pointing it out. µg/mg has been   corrected   to µg/ml.

11. This has been corrected.

12. Necessary changes have been done.

13.Thanks. YAP-1 from line 146 has been removed. Description for Fig. 6 C-E has been changed to match the figure legend.

14. Description for BLM treatment included in Fig. 6A legend.

15. These errors have been corrected in the text.

16. The information on scalebar is now included in the figure legend. The images shown are merged ones. In Fig. 7A, the BLM and TGF- β induction shows YAP1 translocation where nucleus is pinkish whereas in the PF543 treated groups, the nucleus is blue. We have included the non-merged images as supplementary material.

17. This has been corrected.

18. Sorry, we are not able to see any difference as pointed out by the reviewer.

19. The knockdown mentioned in line 234 refers to the in vivo experiments where Sphk1 was deleted in fibroblasts, epithelial and endothelial cells.

20. This is corrected in the revised version.  Figure 11 describes and shows YAP1 translocation to nucleus and its reversal.

21. In the last version, one figure did not get inserted. This has been corrected and schema is Figure 12 as part of the conclusion.

22. Thanks. The font size has been unified in the entire manuscript.

Round 2

Reviewer 1 Report
1. The authors have adressed many of the concerns.
2. However, they feel unable to include whole gel range- stained Western blots of the different proteins. They argue that the short deadline of the submission makes it impossible to include these data. I would like to stress that it is more important to show convincing data for the reader than to keep the deadline and leave the quality of used antibodies unclear.

3. Instead of including new figures in the main manuscript, the authors may also include these data in the supplement.

4. In this view, the blot in Fig. 8B shows fibronectin, that is labelled as a 37 kDa Protein. However, FN has a size of >200 kD.

5. Such incorrect labelling would not occur if the whole range of the gel would be shown, including MW markers. 

Author Response

Manuscript ID: IJMS-718725

Title: Sphingosine kinase 1/S1P signaling contributes to pulmonary fibrosis by activating Hippo/YAP pathway and mitochondrial reactive oxygen species in lung fibroblasts

Authors: Long Huang, Tara Sudhadevi, Panfeng Fu, Prasanth Kumar Punathil-Kannan, David Lenin Ebenezer, Ramaswamy Ramchandran, Vijay Putherickal, Paul Cheresh, Guofei Zhou, Alison Ha, Anantha Harijith, David W. Kamp, Viswanathan Natarajan *

RESPONSE TO REVIEWERS COMMENTS

Reviewer #1:

1. Thanks.

2. As emphasized by the reviewer, we have now provided original images of the relevant bands of our Western blot. In many instances, the blot was cut after transfer based on the molecular weight of the proteins. As an example, in Figure 1, the gel was cut at ~100kDa to blot for fibronectin (~220kDa) and between 25-50kDa for alpha-smooth actin (~42kDa). The same membrane was stripped and probed for GAPDH (~37kDa).

3. We have presented our results sequentially based on the observed data from pre-clinical and in vitro models. We are hoping that the revised versions have significantly improved the manuscript, based on the reviewers’ critiques, wherever possible.

4. This labelling error has been corrected in the revised manuscript.

5. As suggested, the additional blots have been included in the Supplementary File.

Reviewer 2 Report

In the revised manuscript, the authors have satisfactorily addressed all issues I had raised in my original review. Now I believe this manuscript should appeal to the readership of IJMS.

Author Response

Manuscript ID: IJMS-718725

Title: Sphingosine kinase 1/S1P signaling contributes to pulmonary fibrosis by activating Hippo/YAP pathway and mitochondrial reactive oxygen species in lung fibroblasts

Authors: Long Huang, Tara Sudhadevi, Panfeng Fu, Prasanth Kumar Punathil-Kannan, David Lenin Ebenezer, Ramaswamy Ramchandran, Vijay Putherickal, Paul Cheresh, Guofei Zhou, Alison Ha, Anantha Harijith, David W. Kamp, Viswanathan Natarajan *

RESPONSE TO REVIEWERS COMMENTS

Reviewer #2:

No comments

Round 3

Reviewer 1 Report

All concerns are now adequately addressed